SOFTWARE

# G4SNVHunter: An R/Bioconductor Package for Evaluating SNV-Induced Disruption of G-Quadruplex Structures Leveraging the G4Hunter Algorithm

Rongxin Zhang[1,2¤☯], Wenyong Zhu[1☯], Ke Xiao[1], Jean-Louis Mergny[2]*, Xiao Sun[1]*

**1** State Key Laboratory of Digital Medical Engineering, School of Biological Science and Medical Engineering, Southeast University, Nanjing, China, **2** Laboratoire d'Optique et Biosciences (LOB), Ecole Polytechnique, CNRS, INSERM, Institut Polytechnique de Paris, Palaiseau, France

☯ These authors contributed equally to this work.
¤ Current address: Robert Lurie Comprehensive Cancer Center, Department of Obstetrics and Gynecology, Feinberg School of Medicine, Northwestern University, Chicago, Illinois, United States of America
* jean-louis.mergny@polytechnique.edu (J-LM); xsun@seu.edu.cn (XS)

## Abstract

G-quadruplexes (G4s) are nucleic acid secondary structures with important regulatory functions. Single-nucleotide variants (SNVs), one of the most common forms of genetic variation, can potentially impact the formation of G4 structures if they occur within G4 regions. However, there is currently a lack of software tools specifically designed to assess such effects. Here, we present an R/Bioconductor package named G4SNVHunter, which enables rapid detection of variants that may disrupt G4 structures. This tool, based on the core principles of the G4Hunter algorithm, can provide precise quantitative assessment of the propensity for G4 formation within genomic sequences. Specialized experimental methods can then be designed based on the results provided by G4SNVHunter to further verify the specific functions of the affected G4 structures, facilitating deeper insights into the biological impacts of genetic variants from the perspective of G4 structures. To showcase the functionality of the G4SNVHunter package, we analyzed the Neandertal and Denisovan archaic introgressed variants detected by the Sprime software, and identified approximately 5,800 variants located within G4 regions, among which around 230 may impair G4 structure formation propensity. The source code for the G4SNVHunter package has been publicly released under the MIT license at https://github.com/rongxinzh/G4SNVHunter and https://bioconductor.org/packages/devel/bioc/html/G4SNVHunter.html.

## Introduction

G-quadruplexes (G4s) are among the non-canonical secondary structures that typically forms in guanine-rich regions by stacking G-quartets [1], where adjacent

**Data availability statement:** All relevant data are within the manuscript and its Supporting Information files.

**Funding:** This project was supported by the National Natural Science Foundation of China (No. 62472084), the Leading Technology Program of Jiangsu Province (No. BK20222008), and China Scholarship Council (202106090125). J.L.M. thanks recurrent funding from Inserm, CNRS, and Ecole Polytechnique. The funders had no role in the study design, data collection and analysis, decision to publish, or preparation of the manuscript.

**Competing interests:** The authors have declared that no competing interests exist.

guanines in a G-quartet are connected by Hoogsteen hydrogen bonds to create a stable conformation [1]. A classic definition of a G4 sequence involved a pattern such as $G_x N_{1-7} G_x N_{1-7} G_x N_{1-7} G_x$ (where $x$ is generally $\geq 3$ and N can be any base) [2]; however, the actual G4 patterns can be very complex and do not always strictly adhere to this motif [3]. More permissive queries using $x \geq 2$ or longer loops, up to 12 nucleotides, allow to retrieve more candidate sequences but generate a lot of false positives. Therefore, over the past two decades, considerable efforts have been made to improve predictions of G4-prone sequences on a genome-wide scale. These efforts have let to the development of numerous algorithms, with G4Hunter being widely used for its high accuracy [4]. Unlike traditional algorithms, G4Hunter eschews reliance on specific motif patterns for identifying G4-prone regions to avoid overlooking non-canonical G4 sequences capable of forming stable structures.

Growing evidence indicates that G4 structures are neither "junk" nor mere "accidents" in the human genome; instead, they possess critical functions. For example, they are key components of human *cis*-regulatory elements [5], serve as hub nodes for transcription factor binding [6], can even be considered as promoter elements controlling nucleosome exclusion and RNA polymerase II pausing [7], and influence genomic modifications such as DNA methylation [8]. The functionality of G4s is more dependent on the formation of their secondary structures than on their sequence patterns (high guanine content) alone [5]. If these structures cannot form stable conformations within cells, their functions may be directly affected.

Single-nucleotide variants (SNVs), as the most common type of genetic variation, can modulate formation propensity of G4 structures by altering their base composition, either enhancing or impairing G4 stability. Disruptive effects may arise when SNVs impair the formation of G-quartet planes, or increase the cytosine content that leads to Watson-Crick base pairing between cytosine and guanine. Unfortunately, to date, there is no mature software, especially no R package, that can directly assist researchers in evaluating the impact of variants on G4 structure formation. Given the critical regulatory role of G4 structures, variants affecting these structures may lead to functional imbalances. Such effects may help us understand the consequences of certain variants, especially those in non-coding regions.

Here, we introduce an R/Bioconductor package called G4SNVHunter, which is based on the G4Hunter algorithm [4] and specifically designed to identify SNVs, also applicable to other small-scale variants like indels (insertions and deletions), that may impair the formation of G4 structures. To make our package accessible to users with limited programming skills in the G4 field, we have simplified and encapsulated the complex processing steps into user-friendly functions where core operations can be done with just a few basic function calls. In summary, the G4SNVHunter package can provide researchers with a G4-based perspective for variant screening by exploring the possible effects of variants on G4 formation. Based on the output of G4SNVHunter, researchers can design and conduct biological experiments to further explore the mechanism of variants by affecting the formation of G4 structures and thus the biological function.

## Design and implementation

### Principle of the G4Hunter algorithm

The G4Hunter algorithm is a computational method designed to predict G-quadruplex propensity in DNA or RNA. G4Hunter assigns a score to each nucleotide position based on the presence and clustering of guanines (G). The algorithm recognizes that sequences rich in Gs and containing stretches of consecutive guanines are likely to form G4 structures. Each isolated G is assigned a score of +1 while clustered guanines, in runs of 2, 3 or 4 + G get a value of +2, + 3 and +4, respectively. Cytosines (C) are given a symmetrical negative score, from -1 to -4. Adenine (A) and thymine (T) are considered neutral and assigned a score of 0. This algorithm therefore considers G/C skewness as a major factor driving G4 formation.

The G4 Hunter score is then calculated for a given window—we initially performed queries for windows of 20 nucleotides but now recommend a default value of 25. This window size matches the expected length of many G4 structures, usually 25–30 nucleotides, although motifs as short as 14 nucleotides (for the thrombin-binding aptamer), or longer than 40 nucleotides may be stable under physiological conditions. A sliding window (typically of 25 nucleotides) moves across the sequence to calculate the average score within that window. G-rich motifs will give a positive value, and C-rich sequences a negative one (S1 Fig). The algorithm then compares the average score of a window against a predefined threshold. If the score within the window exceeds the threshold, that segment is flagged as a potential G4-forming region (S1 Fig). The threshold can be adjusted to control the sensitivity of detection. We advocate using a threshold of 1.2 or higher for DNA. As RNA G4s tend to be more stable, lower threshold may be tested (*e.g.,* 1.0). Pure AT motifs and random sequences give a score close to 0, meaning that G4 formation is extremely unlikely. Interestingly, G4Hunter score is only weakly correlated with GC content: alternating (CG) or (CCGG) repeats, while being 100%GC, would give a G4Hunter score of 0, with G4 conformation excluded under most conditions.

We developed an optimized version of the G4Hunter search tool by integrating R and C++, achieving improved performance without modifying the core algorithm. Users can invoke the *G4HunterDetect* function to perform genome-wide searching of G4s in a fast manner.

### Workflow of the G4SNVHunter package

G4SNVHunter aims to encapsulate complex internal processes into simple and intuitive functions, ensuring that users from diverse backgrounds can easily get started and use the package effectively. The workflow of G4SNVHunter consists of three main steps (Fig 1A and 1B):

(i)   Importing and processing input data;

(ii)  Evaluating the impact of variants on G4 formation;

(iii) Identifying and visualizing disruptive candidate variants with significant impact.

G4SNVHunter requires users to provide a collection of variants of interest and load them into the environment in the form of a *GRanges* object using the *loadVariant* function. Both Variant Call Format (VCF) and Mutation Annotation Format (MAF) are supported. In addition, the genomic sequences where these variants are located should be provided and imported in the form of a *DNAStringSet* object leveraging the *loadSequence* function. Based on these sequences, the built-in G4Hunter program (via the *G4HunterDetect* function) can be invoked to identify high-confidence G4-prone sequences, with parameters that can be flexibly adjusted. We have developed a fast version of the G4Hunter program, which allows the package to quickly and accurately locate all potential G4 sequences in large genomes, such as the human genome, while using minimal computational resources (*e.g.,* a single CPU core). Next, users can assess the impact of these variants on the formation potential of the identified G4s through the *G4VarImpact* function. In summary,

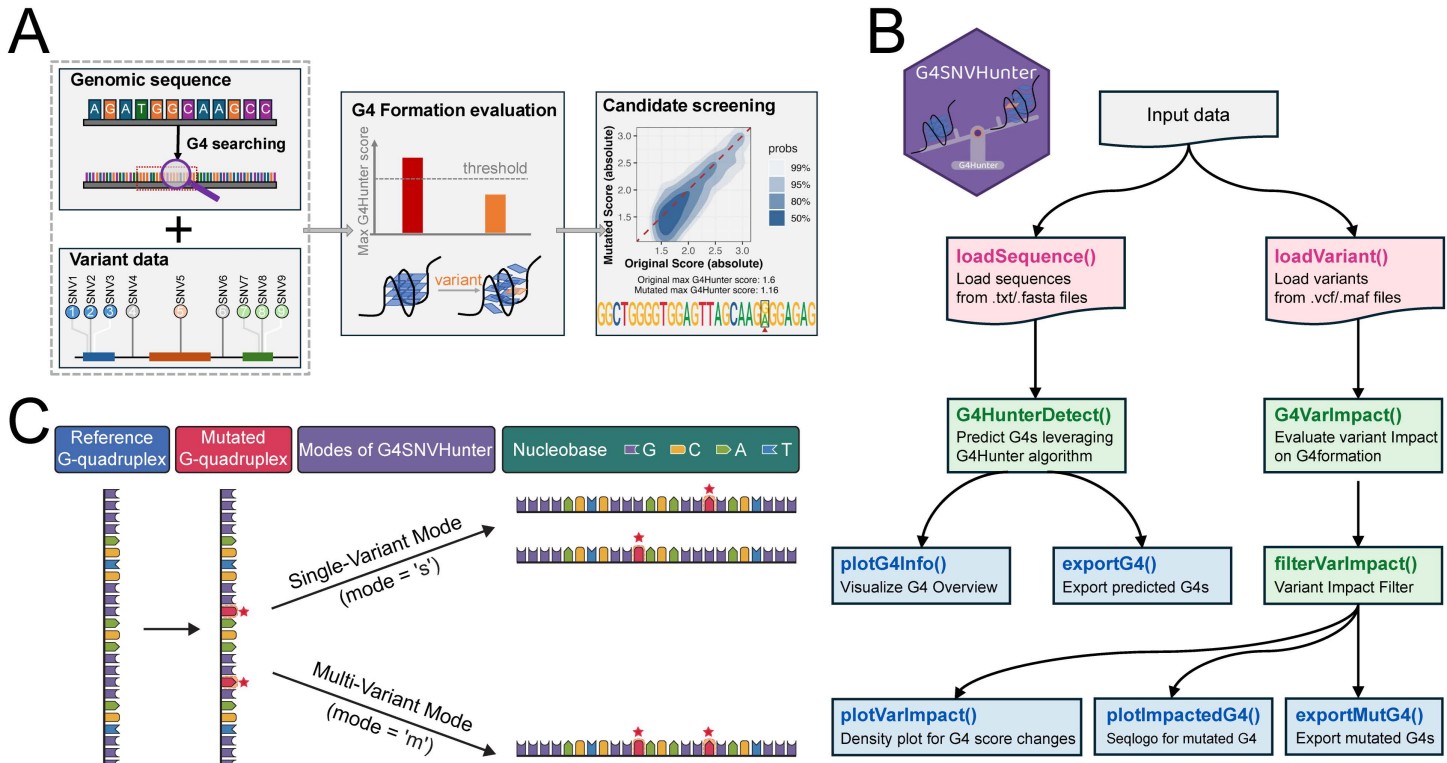

**Fig 1. G4SNVHunter workflow for identifying variants that affect G4 formation. (A)** General workflow for using the G4SNVHunter package. First, the variants and the genomic sequences where these variants are located are provided (Left panel). G4SNVHunter will identify G4-prone regions within these user-provided sequences (Left panel). Subsequently, the impact of the variants on the formation potential of the identified G4s will be assessed based on the G4Hunter algorithm (Middle panel). Finally, candidate variants can be filtered and visualized using functions provided by G4SNVHunter to screen out those that can potentially disrupt the formation of G4 structures (Right panel). **(B)** Function-level schematic of the G4SNVHunter workflow, showing the relationships between key modules and their data flow. **(C)** Two modes for evaluating the impact of variants on G4 formation provided by G4SNVHunter. The first is the single-variant (or variant-centric) evaluation mode, while the second is the multi-variant (or sample-centric) evaluation mode. The difference lies in that the single-variant mode will ignore the sample information and assess each variant individually for its impact on G4 formation. In contrast, the multi-variant mode evaluates the combined effect of multiple variants from a single sample on a given G4 structure. The mutant sites are marked with red stars, assuming they originate from the same sample.

the basic principle of the *G4VarImpact* function is as follows: it first introduces the variant into the original G4 sequence, then rescans the region to compute the maximum G4Hunter score of the mutated sequence, and finally outputs the difference between the original and mutated maximum G4Hunter scores. Here, we provide two assessment modes: single-variant and multi-variant assessments (Fig 1C; see an example in S2 Fig). Both modes involve substituting the corresponding bases for variants within G4 regions, followed by recalculating the G4Hunter score based on these substitutions. However, the single-variant mode assesses each variant individually, making it particularly suitable for scenarios where sample information is less critical, such as analyzing SNP data from the 1000 Genomes Project. The multi-variant (or sample-centric) mode, on the other hand, evaluates the collective impact of variants on G4 formation at the sample level. Unlike the single-variant mode, when multiple variants from the same sample are located within a specific G4, the program will assess their combined impact on that G4. This mode is especially relevant, for instance, when analyzing variants from cancer patients. Since most variants have a minimal impact on G4 formation, further filtering of the results is often necessary. Users can flexibly filter the results using the *filterVarImpact* function to isolate variants that may have a

more disruptive effect on G4 formation (for instance, GGGTGGCGGAGGACGGTCGACG**G**GG with a maximum G4Hunter score of 1.52 to a mutant one GGGTGGCGGAGGACGGTCGACG**A**GG with a maximum G4Hunter score of 1.08 can be identified by G4SNVHunter). Additionally, users can visualize the results using functions such as *plotVarImpact* and *plotImpactedG4*, which intuitively display the location of variants and their scores after variation, thereby providing a clearer understanding of the impact of variants on G4 structures.

## Results

We present an example of archaic introgressed variants here to demonstrate the functionality of the G4SNVHunter package. The data and the analysis script used in the case study of this manuscript are available at https://github.com/rongxinzh/G4SNVHunter-aSNP.

Archaic introgression is a fascinating area that has attracted great attention from scientists. We retrieved Neandertal and Denisovan introgressed variants located on the autosomes—referred to as archaic SNPs (aSNPs)—from the study of Browning *et al* [9]. These archaic variants have flowed into the modern human genome from the genomes of archaic humans. Certain archaic introgressed variants or sequences have been found to be associated with a variety of diseases, traits, and biological functions, including severity of COVID-19 [10], mechanical pain sensitivity [11], circadian rhythms [12], and innate immunity [13]. However, it remains unclear whether these archaic variants can affect the formation propensity of G4 structures. Identifying archaic variants that influence G4 formation propensity would allow experimental biologists to validate through bioassays whether such alterations could affect genomic function, thus providing insights into the mechanisms of these introgressed variants within the framework of G4 structures.

Using G4SNVHunter, we found that over half of the aSNPs (3395 out of 5821) located within G4 regions tend to reduce their formation potential (Fig 2A). Among them, we identified over 230 aSNPs that can potentially affect G4 structure formation by directly destabilizing G4s—from a stable state (|Max G4Hunter Score| ≥ 1.5) to an unstable or less stable one (|Max G4Hunter Score| ≤ 1.2) (Figs 2B and S3, S1 Table). Interestingly, we found that these G4-affecting aSNPs exhibited higher regulatory potential compared to other aSNPs matched in number and nucleotide composition (Fig 2C). By using the VEP online tool (https://grch37.ensembl.org/Tools/VEP) to annotate these aSNPs, we found that most of them are located in intron (40.3%) and intergenic (37.8%) regions (Fig 2D and S2 Table), with a few aSNPs (6.3%) located upstream of genes (within 1,000 bp of the transcription start site). One such aSNP (rs11575044) introgressed from Neandertal and Denisovan genomes is located in the CD7 promoter region. This aSNP significantly reduces the G4 formation potential (maximum G4Hunter score from 1.66 to 1.20) (Fig 2E). CD7 encodes a transmembrane glycoprotein expressed mainly in T cells and natural killer cells [14], which plays an important role in the human immune system and is a potential therapeutic target for hematologic malignancies [15,16]. By querying RegulomeDB, this variant was found to have high regulatory potential (RegulomeDBV2.2 score: 0.98; range: 0–1), suggesting functional relevance. In fact, the impact of Neandertals on the modern human immune system has long been a topic of considerable research interest [17,18]. We here propose a novel possibility based on the G4 perspective, but this hypothesis will require further experimental verification. Notably, we also found an aSNP (rs2073764; https://www.ebi.ac.uk/gwas/variants/rs2073764) located in the intron region of the GNB1L gene, which has been reported to be associated with non-syndromic cleft lip with cleft palate in the Chinese Han population [19]. G4SNVHunter suggests that this variant can disrupt the formation propensity of a G4 structure (S4 Fig). Moreover, the frequency of this introgressed variant is relatively higher in Asian populations compared to other populations, according to the gnomAD and 1 KG databases (S4 Fig). Future studies could explore whether this variant contributes to non-syndromic cleft lip with cleft palate, possibly through its effect on G4 structure formation.

Overall, G4SNVHunter can quickly identify variants that disrupt the formation propensity of G4 structures, thereby offering a structural perspective interpretation of non-coding variants for downstream analysis.

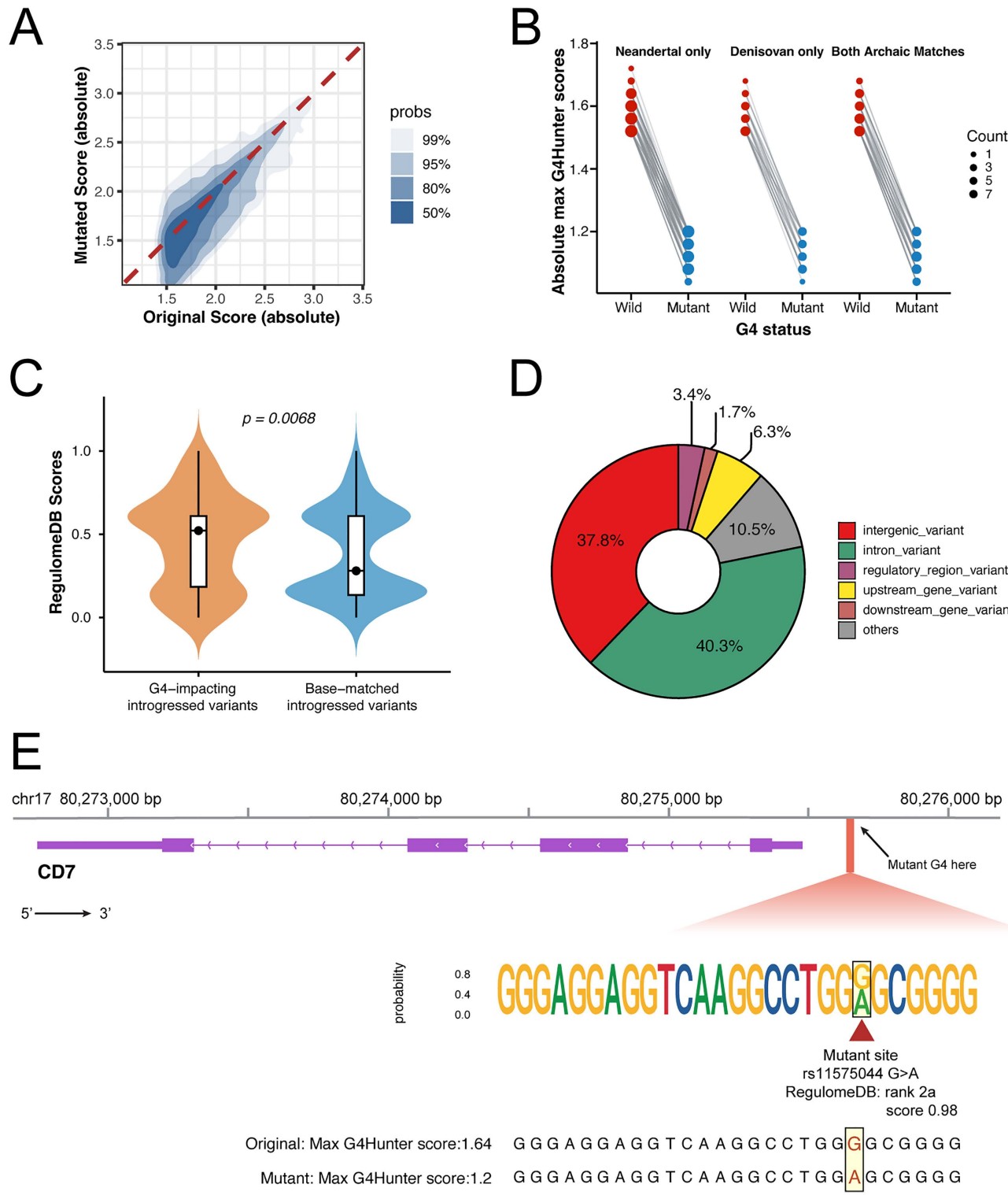

**Fig 2. Overview of archaic SNPs (aSNPs) potentially affecting G4 formation, identified by G4SNVHunter. (A)** 2D density plot of the absolute maximum G4Hunter scores for G4s before and after variation. **(B)** Absolute maximum G4Hunter scores for wild-type G4s and their corresponding aSNP-containing mutants, where the predicted formation propensity is altered. Circle size represents the total number of G4-aSNP pairs. From left to right: introgressed aSNPs found only in Neandertals, only in Denisovans, and in both. **(C)** RegulomeDB scores for G4-impacting aSNPs and a matched

set of randomly selected aSNPs with identical count and base composition. **(D)** Donut chart shows the genomic annotation of the G4-impacting aSNPs as determined by VEP online tool. **(E)** An introgressed aSNP (rs11575044) in the promoter region of CD7 may potentially affect G4 formation propensity. After introducing this variant, the maximum G4Hunter score of the G4 decreases from 1.64 to 1.2.

## Availability and future directions

G4SNVHunter is available on the devel branch of Bioconductor: https://bioconductor.org/packages/devel/bioc/html/G4SNVHunter.html. This package is also regularly updated in the GitHub repository: https://github.com/rongxinzh/G4SNVHunter. A user guide for G4SNVHunter has been published on the Bioconductor platform and can be accessed at https://bioconductor.org/packages/devel/bioc/vignettes/G4SNVHunter/inst/doc/G4SNVHunter.html. The code and data used in the example presented in this manuscript are available in the following GitHub repository: https://github.com/rongxinzh/G4SNVHunter-aSNP.

Interpreting the consequences or deleteriousness of variants has long been a challenging task [20]. G4s are functional structures whose formation and stability can be disrupted by variants, thereby affecting their regulatory functions. Previous studies on variants have often overlooked G4 structures; however, our package fills this gap by offering a G4-focused framework for analyzing these variants, which may provide a novel perspective for elucidating the mechanisms underlying variant effects.

G4SNVHunter should find a variety of applications. For instance, users can utilize this package to investigate whether cancer-associated variants might disrupt G4 structures within promoter or enhancer regions, where G4s often regulate the expression of the target genes associated with these regulatory elements [21,22]. Furthermore, G4SNVHunter is also applicable to other biological questions, such as evaluating patterns of genetic variants and divergences in the evolutionary processes across different species. In the case study of this manuscript, we identified numerous introgressed aSNPs from archaic humans, primarily Neandertals and Denisovans, that may affect the formation of G4 structures in the genome. While the consequences of these introgressed aSNPs are currently unknown, researchers can utilize the output from the G4SNVHunter package to screen for specific introgressed variants they are interested in and employ experimental techniques to pinpoint genes that may be regulated by these aSNP-associated G4s, thus determining the potential impact of these archaic human variants on modern humans.

G4Hunter accuracy, while being better than most algorithms, is not perfect. Experimental confirmation *in vitro* is recommended for motifs with G4Hunter scores below 1.5 (above this value, nearly all sequences tested formed a stable G4 motif) [23]. Higher scores not only indicate nearly certain G4 formation, but also stable structures. Many of the sequences tested with score above 1.8 are thermally stable, and sometimes hardly denatured upon boiling. In contrast, relatively G-poor motifs involving long loops and other structuring elements may involve a local G4 core, which is often very difficult to predict. For this reason, it is often desirable to obtain experimental confirmation *in vitro* or in cells. Of note, G4Hunter currently does not calculate the potentially energetic penalty of opening the double-helix to allow intramolecular folding of the G-rich strand. Efforts are currently made to improve G4Hunter accuracy, based on the collection of experimental evidence for thousands of sequences. A future "G4Hunter 2.0" version would involve slightly different parameters, which are arbitrary and may not necessarily be integers, as in the current version.

Overall, G4SNVHunter opens new avenues for exploring G4 structure associated variants, which can provide strong support for experimental research and helping researchers uncover the complex regulatory mechanisms behind these variants.

## Supporting information

**S1 Fig. Schematic diagram of the G4Hunter scanning principle.** In brief, the program calculates the G4Hunter score within a fixed-length sliding window (e.g., 25 bp). Windows with scores above a threshold (e.g., 1.5) are marked as positive windows. Overlapping positive windows are merged into larger G4 regions, and the maximum score within each region is taken as the maximum window score, reflecting the highest structural formation propensity within the G4 region. (S1_Fig.TIF)

**S2 Fig. A hypothetical example illustrating the differences between the two assessment modes in G4SNVHunter.** The top is an example G4 sequence with a maximum G4Hunter score of 1.52. Suppose there are three possible variants located in the G4 region, originating from sample A (blue) and sample B (green). Variant positions are indicated by stars; mutant nucleotides are shown in black. In the single-variant mode, sample information is ignored; therefore, G4SNVHunter will assess the impact of these three variants on that G4 separately, resulting in three evaluation records, each having the maximum G4Hunter score after the introduce of a single variant into the G4 sequence (G > C: 1.28; A > C: 1.24; G > A: 1.08). In contrast, in the multi-variant mode, G4SNVHunter will make calculations on a sample-by-sample basis. Since both G > C and A > C variants originate from sample A, the program will calculate the combined effect of their impact on the formation of that G4 (maximum G4Hunter score drops to 1.00).
(S2_Fig.TIF)

**S3 Fig. 2D density plot of maximum G4Hunter scores for G4s whose formation propensity is computationally impaired by aSNPs.** These G4s have formation propensity scores > 1.5 before variation and < 1.2 after; G4s with increased or mildly reduced propensity were excluded.
(S3_Fig.TIF)

**S4 Fig. An example of an aSNP affecting G4 formation.** The variant (rs2073764) is located in the intronic region of the GNB1L gene and can reduce the formation score of the G4 on the complementary strand from 1.52 (stable state) to 1.08 (unstable state). This variant was found to be associated with non-syndromic cleft lip with cleft palate in a study based on the Chinese Han population. The bar chart on the right shows the frequency of this variant in Asian, European, African, and American populations in the gnomAD and 1000 Genomes Project databases (data source: https://gnomad.broadinstitute.org/variant/22-19811887-C-T?dataset=gnomad_r4, accessed October 2024).
(S4_Fig.TIF)

**S1 Table. List of aSNPs predicted to disrupt G4 formation propensity.**
(S1_Table.XLSX)

**S2 Table. VEP annotation of the aSNPs listed in S1 Table.**
(S2_Table.XLSX)

## Author contributions

**Conceptualization:** Rongxin Zhang, Jean-Louis Mergny.

**Data curation:** Rongxin Zhang.

**Formal analysis:** Rongxin Zhang.

**Funding acquisition:** Rongxin Zhang, Jean-Louis Mergny, Xiao Sun.

**Investigation:** Rongxin Zhang.

**Methodology:** Rongxin Zhang, Jean-Louis Mergny.

**Software:** Rongxin Zhang.

**Supervision:** Jean-Louis Mergny, Xiao Sun.

**Validation:** Wenyong Zhu.

**Visualization:** Wenyong Zhu.

**Writing – original draft:** Rongxin Zhang.

**Writing – review & editing:** Rongxin Zhang, Ke Xiao, Jean-Louis Mergny, Xiao Sun.

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
