## [Decision Letter · Decision Letter 0]

25 Jul 2025

Dear Dr. Sun,

We are pleased to inform you that your manuscript 'G4SNVHunter: An R/Bioconductor Package for Evaluating SNV-Induced Disruption of G-Quadruplex Structures Leveraging the G4Hunter Algorithm' has been provisionally accepted for publication in PLOS Computational Biology.

Best regards,

Michael Domaratzki

Academic Editor

PLOS Computational Biology

Ilya Ioshikhes

Section Editor

PLOS Computational Biology

Reviewer's Responses to Questions

**Comments to the Authors:**

Reviewer #1: The authors have addressed all my concerns in the revision.

Reviewer #2: The authors address all my comments and concerns.

**Have the authors made all data and (if applicable) computational code underlying the findings in their manuscript fully available?**

Reviewer #1: None

Reviewer #2: Yes

PLOS authors have the option to publish the peer review history of their article (what does this mean? ). If published, this will include your full peer review and any attached files.

**Do you want your identity to be public for this peer review?** For information about this choice, including consent withdrawal, please see our Privacy Policy .

Reviewer #1: No

Reviewer #2: **Yes: ** Alexej Abyzov

---

## [Editor Report · Acceptance letter]

PCOMPBIOL-D-25-01074

G4SNVHunter: An R/Bioconductor Package for Evaluating SNV-Induced Disruption of G-Quadruplex Structures Leveraging the G4Hunter Algorithm

Dear Dr Sun,

I am pleased to inform you that your manuscript has been formally accepted for publication in PLOS Computational Biology. Your manuscript is now with our production department and you will be notified of the publication date in due course.

With kind regards,

Benedek Toth
